# Characterization of Lung Microbiomes in Pneumonic Hu Sheep Using Culture Technique and 16S rRNA Gene Sequencing

**DOI:** 10.3390/ani13172763

**Published:** 2023-08-30

**Authors:** Yongqiang Miao, Xueliang Zhao, Jianlin Lei, Jingru Ding, Hang Feng, Ke Wu, Jiaohu Liu, Chunyang Wang, Dongyang Ye, Xinglong Wang, Juan Wang, Zengqi Yang

**Affiliations:** College of Veterinary Medicine, Northwest A&F University, Yangling 712100, China; 2020065012@nwafu.edu.cn (Y.M.); zhaoxueliang92@sina.com (X.Z.); leijianlin@163.com (J.L.); d1427962909@163.com (J.D.); fh191483594@163.com (H.F.); 15619045817@163.com (K.W.); chunyang@nwafu.edu.cn (C.W.); wxlong@nwsuaf.edu.cn (X.W.); juanwang1234@hotmail.com (J.W.)

**Keywords:** Hu sheep, pneumonia, *Mannheimia*, microbiome, 16SrRNA sequencing

## Abstract

**Simple Summary:**

Pneumonia, which represents a leading cause of mortality among sheep, presents a formidable peril to the sheep industry. In this study, we combine high-throughput 16SrRNA gene sequencing and bacterial culturing to meticulously examine the bacterial community present in lung samples obtained from healthy sheep, those with moderate pneumonia, and those suffering from severe pneumonia. The results demonstrate a significant decrease in pulmonary microbial diversity and alterations in taxonomic composition following the onset of pneumonia. Our results suggest that *M. haemolytica* and *Fusobacterium* are the crucial pathogens in the sheep pneumonia. In addition, the occurrence of sepsis in severe pneumonia sheep could contribute to the eventual mortality outcome. Taken together, the results of this study contribute to guiding preventive and therapeutic measures for pneumonia of different severities.

**Abstract:**

Hu sheep, a locally bred species in China known for its high productivity, is currently suffering from pneumonia. Here, we combine high-throughput 16SrRNA gene sequencing and bacterial culturing to examine the bacterial community in pneumonic Hu Sheep lungs (*p* < 0.05). The results showed that the abundance and diversity of lung bacteria in healthy sheep were significantly higher than those in pneumonia sheep (*p* = 0.139), while there was no significant difference between moderate and severe pneumonia. Furthermore, the composition of the lung microbiota community underwent significant alterations between different levels of pneumonia severity. The application of LEfSe analysis revealed a notable enrichment of *Mannheimiae* within the lungs of sheep afflicted with moderate pneumonia (*p* < 0.01), surpassing the levels observed in their healthy counterparts. Additionally, *Fusobacterium* emerged as the prevailing bacterial group within the lungs of sheep suffering from severe pneumonia. Integrating the results of bacterial isolation and identification, we conclusively determined that *Mannheimia haemolytica* was the primary pathogenic bacterium within the lungs of sheep afflicted with moderate pneumonia. Furthermore, the exacerbation of pneumonia may be attributed to the synergistic interplay between *Fusobacterium* spp. and other bacterial species. Our results provide new insights for guiding preventive and therapeutic measures for pneumonia of different severities in sheep.

## 1. Introduction

Sheep have long held a pivotal position as a vital economic animal worldwide. Nonetheless, pneumonia, which represents a leading cause of mortality among sheep, presents a formidable peril to the sheep industry [1,2]. Currently, *Mannheimia haemolytica* (*M. haemolytica*) and *Pasteurella multocida* (*P. multocida*) are generally recognized as the major causative pathogens of pneumonia in sheep [3,4]. Notably, resistance to tylosin and tetracycline has been observed in a subset of individuals [5]. *Mycoplasma ovipneumoniae* (*M. ovipneumoniae*), originally identified within a sheep flock situated in Queensland, Australia in the year 1972 [6], is now widely recognized as a respiratory pathogen that affects both domestic and wild sheep, as well as goats, worldwide [7,8,9]. Moreover, *Escherichia coli* (*E. coli*) and *Klebsiella pneumoniae* (*K. pneumoniae*) have also emerged as pathogenic bacteria associated with pneumonia in sheep [10]. Recent studies have demonstrated that co-infection with the aforementioned bacteria significantly exacerbates the severity of pneumonia in sheep [11]. Mounting evidence substantiates the notion of a respiratory microbiome, elucidating unequivocal connections between modifications in lung microbiota and the occurrence as well as the intensification of pneumonia [12]. However, the precise connections and dynamic characteristics remain unknown. Hu sheep, a renowned sheep breed in China, stands out for its remarkable characteristics of early maturation and exceptional productivity [13]. There have been a growing number of reported cases of sheep pneumonia in China in recent years [9,14]. Moreover, numerous studies have endeavored to characterize the variability of lung microbiotas in healthy sheep [15,16]. However, the composition of the lung microbiota in Hu sheep affected by pneumonia remains elusive and requires further investigation. Hence, unraveling the characteristics of the lung microbiota in sheep exhibiting diverse levels of pneumonic severity is of paramount importance in devising efficacious strategies for the prevention and control of sheep pneumonia.

The exploration of the human lung microbiome in the context of pulmonary diseases is undergoing continuous expansion, encompassing conditions such as lung cancer, lung adenocarcinoma, and emphysema [17,18]. Sheep have been extensively utilized as translational models in the realm of human lung research, facilitating valuable insights and advancements in our understanding of pulmonary conditions [19,20]. However, there is a relative scarcity of data on the microbiome of sheep lung diseases [21]. Therefore, conducting investigations into the lung microbiome of sheep assumes great significance not only for understanding sheep lung diseases, but also for the study of human lung diseases.

The objective of this study was to characterize the viable bacterial community in lung samples from healthy sheep, moderate pneumonia sheep, and severe pneumonia sheep and to determine the main pathogens. We hope this research will provide insights into the prevention and treatment of pneumonia in Hu sheep.

## 2. Materials and Methods

### 2.1. Animals and Sample Collection

An outbreak of pneumonia recently unfolded at a significant Hu sheep farm situated in Yulin (109°75′ N, 38°28′ E), Shaanxi province. Based on meticulous statistics compiled within this farm, pneumonia emerged as the primary cause of mortality among fattening sheep during March 2022. The main clinical presentation was respiratory distress, runny nose, wet cough, and hyperthermia. Eye, nasal, cough and respiratory score were classified referenced the Wisconsin sheep health scoring criteria https://www.vetmed.wisc.edu/dms/fapm/fapmtools/8sheep/sheep_respiratory_scoring_chart.pdf/ (accessed on 15 April 2022). According to the health scoring criteria, a total of 18 Hu sheep (aged 4–5 months) were selected from the farm for the purpose of this experiment in April 2022. The sheep cohort comprised six individuals classified as healthy (group H), six individuals afflicted with moderate pneumonia but without any therapeutic intervention (group M), and an additional six individuals suffering from severe pneumonia (group S), also without therapy. The details about the sheep are listed in Appendix A. All fattening sheep were under the same feeding procedures and management conditions.

### 2.2. Sample Collection

Blood samples were obtained aseptically from 18 sheep via the jugular vein and anticoagulated with EDTA for bacterial culture purposes. Subsequently, all the sheep were euthanized in a humane manner, and their thoracic cavities were meticulously opened to expose the lungs. During this process, careful observation and documentation were conducted to assess the presence of lung–chest adhesions. A detailed description of the lung condition observed following slaughter in both clinically healthy and pneumonia-affected Hu sheep can be found in Appendix A. Finally, the separated whole lungs were photographed against a clean background. To ensure the acquisition of a representative sample, we promptly extracted 1 g of lung tissue from each of the seven lobes of every lung (shown in Appendix A). These tissue samples were rapidly mixed, transferred into 50 mL centrifuge tubes, and immediately frozen in dry ice. Subsequently, the frozen samples were transported to our laboratory and stored at a temperature of −80 °C until further analysis. Additionally, the blood samples were cryopreserved and transferred to our laboratory to facilitate the subsequent isolation of bacteria.

### 2.3. Microbial Isolation and Identification

The frozen lung tissues were meticulously homogenized on ice utilizing a specialized homogenizer, ensuring thorough and consistent tissue disruption. Both the blood and lung homogenate were streaked onto the brain heart infusion agar medium containing 5% sheep blood, followed by incubation at 37 °C in both aerobic and anaerobic conditions to monitor any bacterial growth. The dominant colonies were picked for purification culture. Identification was performed as previously described [14]. The genomic DNA of isolates were extracted by a genomic DNA purification kit (DAKEWE, Shenzhen, China) according to manufacturer recommendations. The universal primers 27F (forward primer 5′-AGAGTTTGATCCTGGCTCAG-3′) and 1492R (reverse primer 5′-GGTTACCTTGTTACGACTT-3′) were used to amplify the isolates’ 16S rDNA. The PCR product was sent for sequencing (Tsingke, Xi’an, China) and the results were analyzed using the NCBI BLAST (https://blast.ncbi.nlm.nih.gov/ (accessed on 23 April 2022)) algorithm for homologous sequence searches.

### 2.4. DNA Extraction, PCR and Sequencing

Frozen lung tissues were homogenized on ice using a homogenizer and the DNA was extracted using Mabio DNB361B Bacterial DNA Extraction Mini Kit (Mabio, Guangdong, China) according to the manufacturer’s instructions. The V4–V5 region of the 16S rRNA gene was amplified with barcode-indexed universal bacterial primers (forward primer 515F, 5′-GTGCCAGCMGCCGCGGTAA-3′ and reverse primer 806R, 5′-CCGTCAATTCMTTTRAGTTT-3′ [22]. PCR reactions, containing 25 μL 2× Premix Taq (Takara Biotechnology, Co., Ltd., Dalian, China), 1 μL each primer (10 μM) and 3 μL DNA (20 ng/μL) template in a volume of 50 µL, were amplified by thermocycling: 5 min at 94 °C for initialization; 30 cycles of 30 s denaturation at 94 °C, 30 s annealing at 52 °C, and 30 s extension at 72 °C; followed by 10 min final elongation at 72 °C. The PCR instrument was a BioRad S1000 (Bio-Rad Laboratory, Hercules, CA, USA). The length and concentration of the PCR product were detected using 1% agarose gel electrophoresis. Samples with a bright main strip between (V4–V5: 400–450 bp) could be used for further experiments. PCR products were mixed in equidensity ratios using the GeneTools Analysis Software (Version4.03.05.0, SynGene, Cambridge, UK). Then, the mixture of PCR products was purified with E.Z.N.A. Gel Extraction.

Sequencing libraries were generated using NEBNext^®^ Ultra™ II DNA Library Prep Kit for Illumina^®^ (New England Biolabs, Ipswich, MA, USA) following vmanufacturer’s recommendations, and index codes were added. The library quality was assessed on the Qubit@ 2.0 Fluorometer (Thermo Fisher Scientific, Waltham, MA, USA). Finally, the library was sequenced on an Illumina Nova6000 platform and 250 bp paired-end reads were generated (Guangdong Magigene Biotechnology Co., Ltd., Guangzhou, China).

### 2.5. Sequence Analysis

Fastp (version 0.14.1, https://github.com/OpenGene/fastp/ (accessed on 23 April 2022) was used to control the quality of the raw data using a sliding window (-W 4 -M 20). The primers were removed by using cutadapt software (https://github.com/marcelm/cutadapt/ (accessed on 25 May 2022) according to the primer information at the beginning and end of the sequence to obtain the paired-end clean reads. Paired-end clean reads were merged using usearch-fastq_mergepairs (V10, http://www.drive5.com/usearch/ (accessed on 25 May 2022) according to the relationship of the overlap between the paired-end reads, when there was at least 16 bp overlap with the read generated from the opposite end of the same DNA fragment; the maximum mismatch allowed in the overlap region was 5 bp, and the spliced sequences were called raw tags. Fastp (version 0.14.1, https://github.com/OpenGene/fastp/ (accessed on 25 May 2022) was used to control the quality of the raw data using a sliding window (-W 4 -M 20) to obtain the paired-end clean tags. Sequence analysis was performed using Qiime2 software [23], with which representative sequences were established as operational taxonomic units (OTUs) and aligned using the DEBLUR program integrated within QIIME2.

### 2.6. Statistical Analysis

Bioinformatic analysis of the lung microbiota was carried out using the MAGICHAND Cloud platform (http://cloud.magigene.com/ (accessed on 27 May 2022). Based on the OTU information, rarefaction curves and alpha diversity indices, including observed OTUs, Chao1 richness, and Shannon index, were calculated using usearch-alpha_div (V10, http://www.drive5.com/usearch/ (accessed on 27 May 2022). The similarity among the microbial communities in different samples was determined by principal coordinate analysis (PCoA) and non-metric multidimensional scaling analysis (NMDS) based on Bray-Curtis dissimilarity using the Vegan package.

The linear discriminant analysis (LDA) effect size (LEfSe) (http://huttenhower.sph.harvard.edu/LEfSe/ (accessed on 27 May 2022) was determined to identify the significantly abundant taxa (phylum to genera) of bacteria among the different groups (LDA score = 3). Random forest analysis was performed based on the OTU_table, using the randomforest software package in R software to build prediction mode.

## 3. Results

### 3.1. Outcome of Bacterial Isolation

In total, lung tissue and blood samples from 18 sheep were used for bacterial isolation, and the outcomes of the separation are presented in Figure 1. Overall, 22 bacterial isolates were recovered in the lungs, including seven isolates of *M. haemolytica* (38.89% (7/18)), three isolates of *P. multocida* (16.67% (3/18)), five isolates of *Streptococcus lutetiensis (S. lutetiensis)* (27.78% (5/18)), two isolates of *E. coli* (11.11% (2/18)), three isolates of *Streptococcus pasteuaianus* (*S. pasteuaianus*) (11.11% (2/18)), one isolate of *P. aeruginasa* (5.55% (1/18)), and one isolate of *Streptococcus vicugnse* (*S. vicugnse*) (5.55% (1/18)). We found that bacteria could be isolated in the lungs of both healthy (50.00% (3/6)) and diseased sheep (100% (12/12)). *M. haemolytica* was the dominant species isolated from the sheep with moderate pneumonia. However, a greater number of bacterial species and coinfections were found in lungs from severe pneumonia sheep. It is worth noting that five isolates were only recovered in blood from six sheep with severe pneumonia (83.33% (5/6)), including *P. aeruginasa* (60.00% (3/5)), *Aeromonas veronii* (*A*. *veronii*) (20.00% (1/5)), and *Pseudomonas azotoformans* (*P*. *azotoformans*) (20.00% (1/5)). The sepsis observed in severe pneumonia-afflicted sheep may ultimately contribute to their mortality.

### 3.2. 16S rRNA Sequencing Analysis

After performing filtering, denoising, chimera checking, and singleton checking, we obtained a total of 1,193,756 pairs of 420 bp clean reads. A total of 3049 operational taxonomic units (OTUs) were obtained via clustering with a 97% identity threshold.

From the richness presented in Figure 2A, it can be observed that as the proportion of extracted sequences increases, the richness of the lung bacteria initially increases and then levels off. This indicates that with increasing numbers of sequences in the samples, the detection of bacterial species in the lung gradually stabilizes. Even with continued increase in the number of sample sequences, the possibility of discovering new lung microbial species becomes limited. Therefore, the sequencing data volume obtained in this experiment is sufficiently large to capture the majority of microbial information present in the samples, making it suitable for subsequent analysis.

The Venn diagrams for shared and unique operational taxonomic unit distribution of OTUs and the quantity of OTUs in each sample are presented in Figure 2B,C. The Venn diagrams reveal that a total of 766 OTUs were shared among the three groups, with each group possessing unique OTUs excluded from the others.

### 3.3. Analysis of Microbial Diversity in the Lungs

#### 3.3.1. Bacterial Community α-Diversity

The relative abundance (Chao1) and diversity (Shannon) index values of bacterial communities for each group are compared in Figure 3A,B. The Chao1 and Shannon indices reveal that the bacterial community abundances and diversity in group M and group S were significantly lower than those in group H (*p* < 0.05), while no significant differences were observed in group M and group S.

#### 3.3.2. Bacterial Community β-Diversity

To account for variations in the composition of lung microbial communities across different groups of samples, PCoA based on bray_curtis distances revealed that group H was distinct from group M and group S (Figure 3C) (*p* < 0.05, for details see Appendix A). The NMDS analysis based on bray_curtis demonstrated that group H exhibited significant separation from both group M and group S (Figure 3D) (*p* < 0.05; for details, see Appendix A).

### 3.4. Bacterial Community Composition at Various Taxonomic Levels in the Lungs

It is now recognized that healthy lung, previously believed to be sterile, actually harbor a diverse microbiota. Here, the composition of lung microbial communities in both healthy sheep and those with varying degrees of pneumonia was assessed at different taxonomic levels. The lungs from sheep with moderate or severe respiratory symptoms exhibited bacterial community profiles that were similar to those of asymptomatic sheep. At the phylum level, *Proteobacteria* (30.02%, 75.58%, 16.46%), *Bacteroidetes* (29.83%, 7.60%, 28.61%), *Fusobacteria* (14.24, 6.77, 44.58%), *Firmicutes* (18.90%, 4.21%, 5.96%), and *Tenericutes* (2.51%, 4.91%, 2.85%) were predominant in the healthy, moderate, and severe pneumonia sheep (Figure 4A). At the family level, *Pasteurellaceae* (26.35%, 73.79%, and 14.81%), *Fusobacteriaceae* (14.24, 6.77, and 44.57%), *Bactereoidaceae* (8.35%, 1.71%, and 19.71%), *Prevotellaceae1* (12.61%, 1.43%, and 1.69%), and *Mycoplasmataceae* (2.42%, 4.82%, and 2.78%) were the predominant bacteria in both groups (Figure 4B). At the genus level, *Mannheimia* (12.68%, 60.54%, and 11.58%), *Fusobacterium* (14.24%, 6.77%, and 44.58%), *Bactereoides* (8.35%, 1.71%, and 19.71%), *Pasteurella* (12.07%, 7.75%. and 2.37%), *Prevotella*_1 (7.94%, 1.05%, and 1.12%), and *Mycoplasma* (2.42%, 4.82%, and 2.78%) were the predominant bacteria in both groups (Figure 4C). The distribution and variability of the bacterial genera in the three groups is also displayed in the heatmap shown in Figure 5.

### 3.5. Overview and Comparison of Lung Microbiota at the Genus Level

Previous research has unequivocally demonstrated discernible dissimilarities in both the richness and abundance of lung microbiota contingent upon alterations in health status [17]. A total of 65 different taxa were identified at the genus level in the three groups. Among them, only 10 taxa-*Mannheimia* (*p* < 0.01), *Fusobacterium* (*p* < 0.01), *Bacteroides* (*p* < 0.01), *Prevotella*_7 (*p* < 0.05), *Succiniclasticum* (*p* < 0.05), Bibersteinia (*p* < 0.01), *Pseudomonas* (*p* < 0.05), *Peptoniphilus* (*p* < 0.05), Escherichia-*Shigella* (*p* < 0.01), and *Mogibacterium* (*p* < 0.05)—remained significantly different after applying one-way ANOVA (details shown in Appendix A).

The common bacteria documented in the rumen and gut [24], such as Succiniclasticum, Rikenellaceae_RC9_gut_group, Ruminococcaceae_UCG-005, Lachnospiraceae_NK3A20_group, Christensenellaceae_R-7_group, Prevotellaceae_UCG-001, Ruminococcaceae_UCG-014, and Ruminococcaceae_UCG-002, were identified in both healthy and diseased lungs.

In order to delve into the variations in the sheep lung microbiota among different severities of pneumonia, we performed a difference analysis of group H and group M, and group H and group S at the genus level using Student’s *t* test (Figure 6). As shown in Figure 6A, there were no obvious differences in other lung microbiota between the healthy and moderate groups except for *Manneimia* (*p* < 0.01), *Prevotella*_1 (*p* < 0.05), and *Bacillus* (*p* < 0.05). The results shown in Figure 6B describe the differences in the lung microbiota between healthy sheep and sheep with severely symptomatic pneumonia. *Manheimia*, *Fusobacterium*, and *Bacteroides* also had a high abundance in both the H and S groups. However, the abundance of *Fusobacterium* was significantly higher (*p* < 0.05) in the S group compared to the H group, while *Pasterurella*, *Prevotella_1*, *Bacillus*, *Prevotella_7* was significantly higher (*p* < 0.05) in group H. It is widely believed that mycoplasma infection is more commonly associated with sheep and goat pneumonia; however, in this study, *Mycoplasma* exhibited an increasing tendency in both group M and group S, albeit with insignificant differences.

### 3.6. Comparison of Microbiotas in Lungs

We employed linear discriminant analysis (LDA) effect size (LEfSe) to identify taxa exhibiting differential abundances between the two groups, enabling comparison of their relative contributions. LEfSe analysis demonstrated several features that were over-represented in sheep with asymptomatic, moderately symptomatic, and severely symptomatic pneumonia. As expected, *Prevotellaceae*, one of the major constituents in the healthy individual lower respiratory tract microbiota [12,25], were enriched in the healthy control group. *Fusobacteriaceae*, *Aeromonadaceae*, *Peptococcus* and *Bacteroidaceae* were highly abundant in sheep with severely symptomatic pneumonia. However, in the lungs from sheep with moderately symptomatic, features such as *Pasteurellaceae*, *Mannheimia*, *Actinobacillus*, *Bibersteinia*, and *Escherichia* were more abundant (Figure 7A,B).

### 3.7. Correlation of Lung Microbiota in Sheep with Severity of Pneumonia

Random forest algorithms have been utilized to identify the microbial species/genera that have the most significant impact on diseases [26], while filtering out those with lesser relevance. ‘Mean Decrease Accuracy’ refers to the reduction in accuracy of the random forest prediction, with higher values indicating variables of greater importance. On the other hand, ‘Mean Decrease Gini’ measures how each variable affects heterogeneity at each node of the classification tree, allowing for a comparison of variable importance. Larger values indicate greater significance. Here, the random forest model incorporated the relative abundances of the top 500 bacterial features, and Figure 8 displays the top 15 bacterial features (in genus level) that exhibited the strongest predictive power for outcomes. *Fusobacterium*, *Mannheimia*, *Haemophilus*, and *Escherichia-Shigella* were identified as the most discriminating predictors based on the mean decrease in the Gini criterion (Figure 8A). *Mannheimia*, *Haemophilus*, *Fusobacterium*, and *Escherichia-Shigella* were identified as the most discriminating predictors based on the mean decrease in accuracy (Figure 8B). At the species level, *Mannhiemia_haemolytica* and *Fusobacterium_nucrophorum_subsp._nucrophorum* were identified as the most discriminating predictors based on both the mean decrease in Gini and mean decrease in accuracy criteria (shown in Appendix A). Taken together, these results suggest that *Mannheimia* and *Fusobacterium* are the primary etiological agents responsible for the occurrence and exacerbation of pneumonia in sheep.

## 4. Discussion

The presence of bacteria in the lower respiratory tract has long been acknowledged as a pathological manifestation. However, with the advent of advanced microbiome sequencing technologies, it has come to light that, even in the absence of apparent respiratory diseases, the lungs of healthy individuals harbor a diverse array of microorganisms. Recently, there has been a significant increase in research aimed at understanding the role of the lung microbiota in humans and animal models, particularly in relation to respiratory disease [15]. Indeed, bacterial pneumonia is common in cattle and sheep, and is often associated with high morbidity and mortality. In this study, bacterial communities in the lungs of Hu sheep with pneumonia of varying degrees of severity were studied using 16S rRNA sequencing and culture techniques. The ultimate objective of this study was to identify crucial therapeutic features that could significantly impact clinical outcomes.

Direct sequencing of lung tissue can provide more direct information on the pulmonary microbiome [27]. Recent findings support the designation of whole lung tissue as the most suitable specimen type for conducting comprehensive investigations into the mouse lung microbiome [28]. Hence, we opted to utilize lung tissue as the primary sample for conducting diversity sequencing. Empirical evidence has shown that elevated diversity and richness of microbial species convers favorable effects on the respiratory health status of pigs [29]. In the present study, a reduction in lung microbiome richness and diversity was observed in sheep afflicted with pneumonia, as evidenced by the analysis of Chao values and Shannon indices. This finding is consistent with previous reports in humans, too [30,31]. However, it is noteworthy that no significant disparities were detected in terms of microbial richness and diversity between sheep affected by moderate and severe pneumonia. This finding aligns with the results of previous studies on the human lung microbiome, specifically in patients with stable and exacerbated chronic obstructive pulmonary disease (COPD) [32]. The PCoA and NMDS analyses revealed that the lung microbiota of healthy sheep differed from that of those with pneumonia (moderate and severe). Given that all the sheep shared the same environment (enclosure, drinking water, feed), we postulate that the onset of pneumonia is closely linked to perturbations in the lung microbiota. We hypothesize that in healthy individuals, the lung microbiome maintains a dynamic equilibrium between the influx and elimination of microorganisms. However, when this delicate balance is disrupted by factors such as acute infection with new microorganisms or immune dysfunction, the population of dominant pathogenic bacteria increases, subsequently contributing to the initiation and progression of pneumonia.

This study demonstrated that *Proteobacteria*, *Bacteroides*, *Fusobacteria*, *Firmicutes*, and *Parasitcutes* were the predominant bacteria in the lungs of healthy and diseased sheep, which is consistent with previous findings in human and pig lung microbiomes. These microorganisms may contribute to maintaining pulmonary homeostasis [33,34]. Even though the dominant phyla composition was not altered, its absolute abundance changed significantly. We observed that the proportions of *Proteobacteria* increased in the lungs of sheep with moderately symptomatic pneumonia, while *Fusobacteria* were enriched in those with severely symptomatic pneumonia, and the predominant bacteria in the lungs of healthy sheep were *Firmicutes*. It has been shown that *Firmicutes* encompass a multitude of significant beneficial bacteria, which promote intestinal homeostasis, enhance the immune function, and improve animal growth performance [35]. Whether *Firmicutes* also maintain pulmonary homeostasis requires further study. In this study, *Prevoyella*_*1*, *Prevoyella*_7, *Bacillus*, *Fibrobacter Acinetobacter*, *Ruminoccaceae UCG*-*005*, *Christensenellaceae*_*R7*_*group* were enriched in the lungs of healthy sheep. *Prevotella* and *Ruminococcus* have exhibited negative correlations with lung lesions in pigs [34]. The above study is consistent with our research, indicating their crucial roles in establishing and maintaining a healthy micro-ecological balance.

At the genus level, there is a significant increase in the abundance of *Mannheimia* in the lungs of sheep with moderately symptomatic pneumonia compared to healthy sheep, whereas *Fusobacterium* is significant increased in the lungs of sheep displaying severe pneumonia. A previous study also showed that *Fusobacterium* is one of the most abundant bacteria genera in the lungs and lymph nodes of calves who have died from BRD [36]. Sudarvili et al. showed that *Fusobacterium leukotoxin* is highly toxic for Bighorn Sheep leukocytes, and *F. necrophorum* invades the lungs subsequent to the entry of *M*. *haemolytica* and other aerobic respiratory pathogens, which initiate tissue damage [37]. This trend is highly consistent with the trend observed in our research. We believe that *Mannheimia* and *Fusobacterium* are responsible for pneumonia development and exacerbation. Random forest algorithms based on the genus level further confirmed that *Mannheimia* and *Fusobacterium* were the primary etiological agents responsible for the occurrence and exacerbation of pneumonia in sheep. Therefore, we propose that treatment strategies be individualized based on the severity of the underlying pneumonia.

*Mycoplasma sp*. and *M. haemolytica* are common microorganisms found in the respiratory tract of healthy ruminant animals [11,38], and their pathogenicity is induced under the provocation of various stimuli. Moreover, coinfection of *Mycoplasma* and *M. haemolytica* exacerbates the severity of pneumonia [11]. In our study, we observed higher levels of *Mycoplasma* in the lungs of Hu sheep with moderate and severe pneumonia compared to healthy sheep; however, the difference was not statistically significant.

Robert et al. [39] showed that the lung microbiome exhibits an enrichment of gut bacteria in both murine models of sepsis and humans with established acute respiratory distress syndrome (ARDS). In humans, the upper respiratory tract microbiota is dispersed and colonizes the lung through respiration and microaspiration [40]. In ruminants, the act of rumination further increases the likelihood of rumen flora entering the lungs, which may have an impact on lung microbiota [15]. In this study, the common bacteria documented in the rumen were identified in both healthy and diseased lungs, which is consistent with Glendinning et al. [15]. Nonetheless, the abundance of *Succiniclasticum* in the lungs of healthy sheep exhibited a marked and statistically significant increase compared to those of afflicted individuals (*p* < 0.05). Previous studies have revealed the specialized metabolic capability of *Succiniclasticum* to ferment succinate and convert it into propionate, a process that exerts advantageous effects on gut chemistry [41]. Intranasal bacterial therapeutics have the potential to mitigate *M. haemolytica* colonization in dairy calves [42,43]. Therefore, the role of *Succiniclasticum* in sheep lung homeostasis and the potential of intranasal bacterial therapeutics warrants further investigation. D. Cid et al. [44] pointed out that the association between *P. multocida* isolation and pneumonic lesions was stronger than that of *M. haemolytica*, and the isolation of *P. multocida* resulted in more severe pneumonic lesions compared to *M. haemolytica* [5]. It should be noted that this statistical result was obtained using bacterial culture techniques. 16S rRNA gene sequencing is only accurate until the genus level of taxonomy. Meanwhile, the bacterial culture technique provides a basis for bacterial species identification and for further analysis. In our study, neither isolation nor sequencing techniques were sufficient to explain the association between *P. multocida* and the severity of pneumonia. Consistent with the trends observed in the 16SrRNA diversity sequencing results, the bacterial isolation outcomes also demonstrated a higher prevalence of *M. haemolytica*, particularly in the lungs of sheep with moderate pneumonia. Regrettably, there were no *Fusobacterium spp* that could be isolated from the lungs of all of the sheep. It is necessary to improve the isolation method of *Fusobacterium*. Finally, but most importantly, *A. veronii*, *P. aeruginosa* and *P. azotoformans* were frequently isolated from the bloodstream of sheep suffering from severe pneumonia, which could ultimately be one of the factors leading to their death.

However, there are several limitations that should be acknowledged in the present study. Firstly, the sample size was too small. Secondly, this study did not include healthy sheep from alternative breeding environments for comparative analysis, despite previous research highlighting variations in lung microbiota across different feeding environments. Lastly, it is important to acknowledge the imperfections in the isolation method for strict anaerobic bacteria employed in this study; regrettably, no isolates of *Fusobacterium* spp. were obtained as a result. The next studies in our laboratory will address these issues.

## 5. Conclusions

We combined high-throughput 16SrRNA gene sequencing and bacterial culturing to characterize the viable bacterial community in lung samples. The results demonstrated a significant decrease in pulmonary microbial diversity and alterations in taxonomic composition following the onset of pneumonia. *M. haemolytica* and *Fusobacterium* were identified as the crucial pathogens in sheep pneumonia. In addition, *P. aeruginosa*, *A. veronii*, and *P. azotoformans* were isolated from blood samples in severe pneumonia sheep (83.3% (5/6)). Thus, the occurrence of sepsis in severe pneumonia sheep could contribute to the eventual mortality outcome. Veterinarians need to take sepsis into account when developing a treatment plan. This study contributes to the guidance of preventive and therapeutic measures for pneumonia of different severities.

## Figures and Tables

**Figure 1 animals-13-02763-f001:**
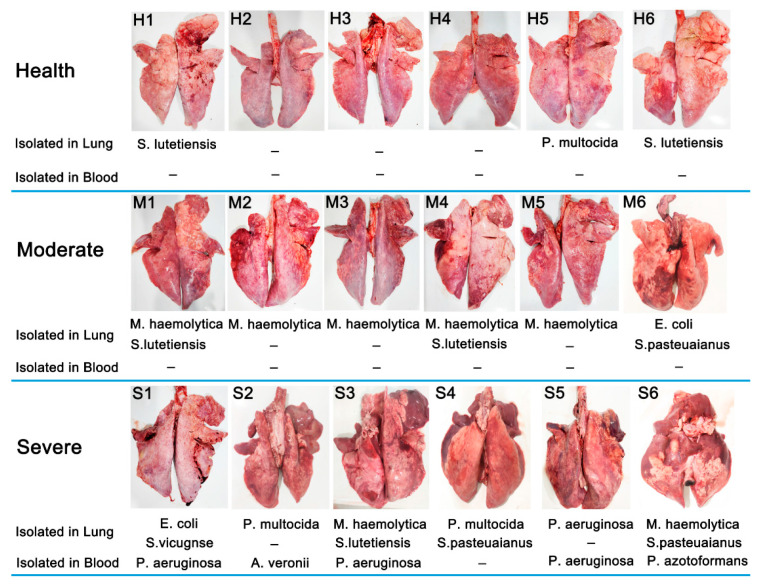
Appearance of the lungs and bacterial isolation results.

**Figure 2 animals-13-02763-f002:**
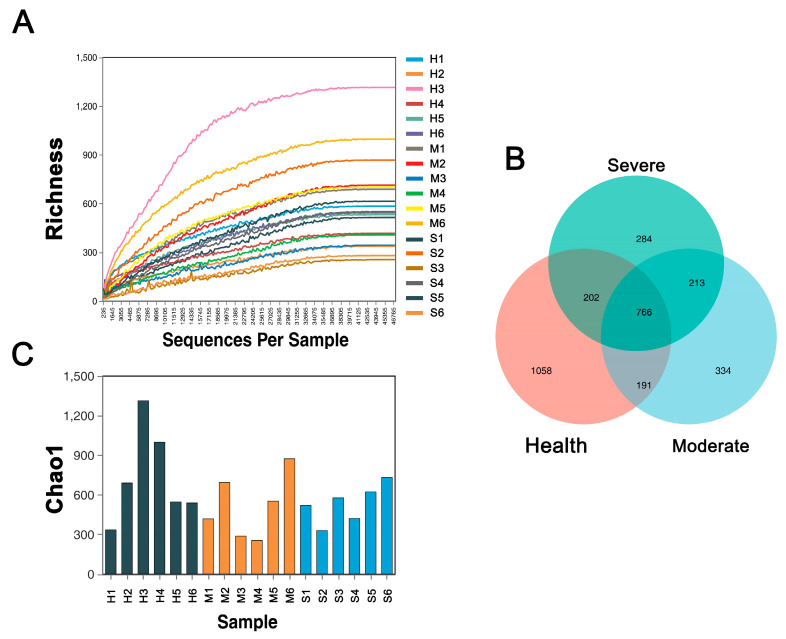
Feasibility analysis of sequencing data. Sequencing depth and evenness of gut microbiota were assessed with (**A**) dilution curves, (**B**) Venn diagrams for shared and unique operational taxonomic unit distribution in OTU, and (**C**) quantity of OTUs in each sample.

**Figure 3 animals-13-02763-f003:**
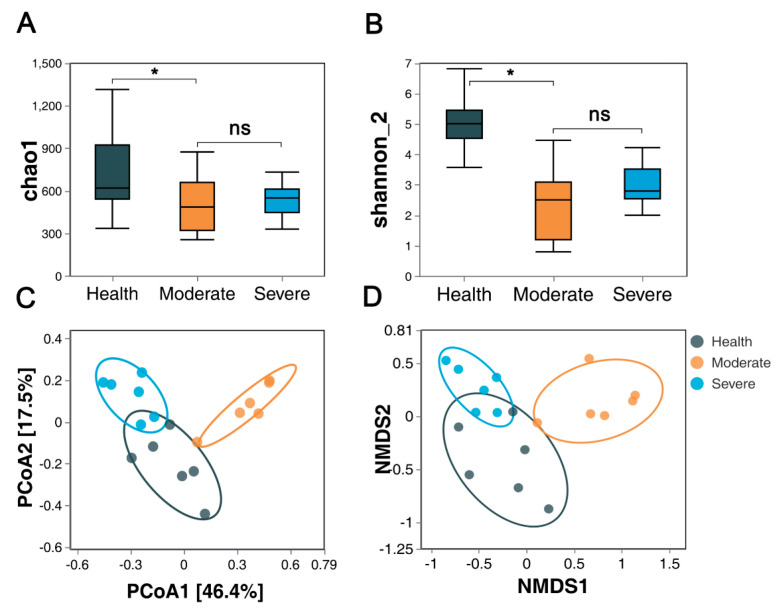
Bacterial community αdiversity (**A**,**B**) and β-diversity (**C**,**D**); (**A**) Chao1 index; (**B**) Shannon diversity index; (**C**) principal coordinates analysis (PCoA) on the OTU level based on bray_curtis; (**D**) non-metric multidimensional scaling analysis (NMDS) on the OTU level based on bray_curtis (* *p* < 0.05).

**Figure 4 animals-13-02763-f004:**
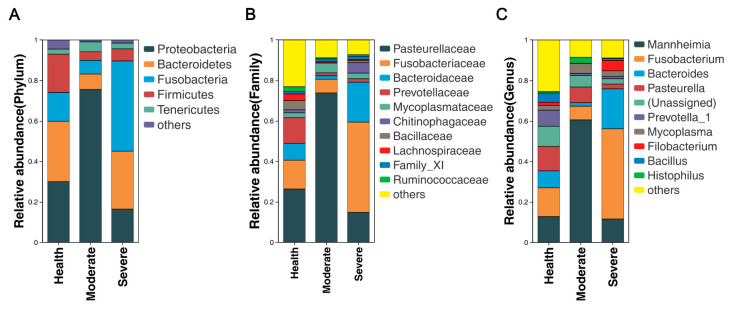
Microbial community bar plot of the top 10 genera at different taxonomic levels in lungs (**A**) at the phylum level (relative abundance >1%), (**B**) at the family level (relative abundance >1%), and (**C**) at the genus level (relative abundance >1%). Values are expressed as mean values.

**Figure 5 animals-13-02763-f005:**
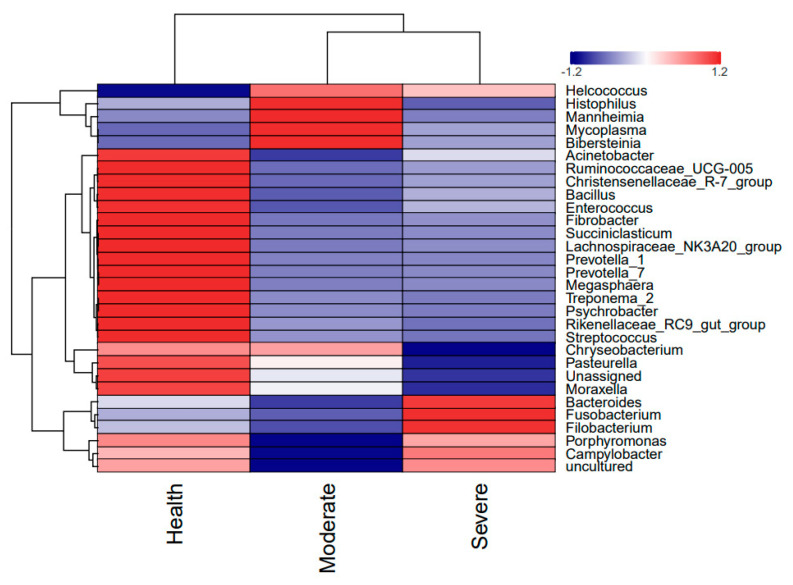
Heatmap of the distribution and variability of the bacteria in the lungs.

**Figure 6 animals-13-02763-f006:**
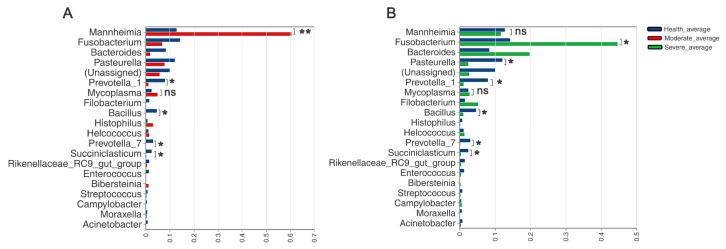
Comparison of lung microbiota between the two groups at the genus level using Student’s *t* test. (**A**) Differences in species abundance between group H and group M; (**B**) differences in species abundance between group H and group S (* *p* < 0.05, ** *p* < 0.01).

**Figure 7 animals-13-02763-f007:**
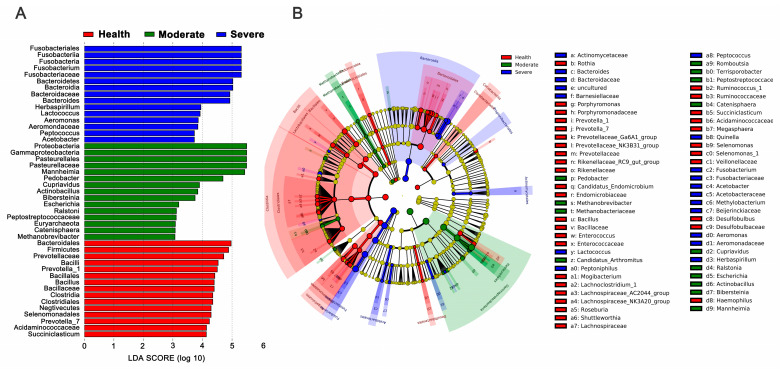
Differential biomarkers in the lung microbiota of sheep associated with asymptomatic, moderately symptomatic, and severely symptomatic pneumonia. (**A**) The significance criterion was set at LDA score >3. (**B**) The cladogram was utilized to visualize the phylogenetic distribution of taxa exhibiting significant differences.

**Figure 8 animals-13-02763-f008:**
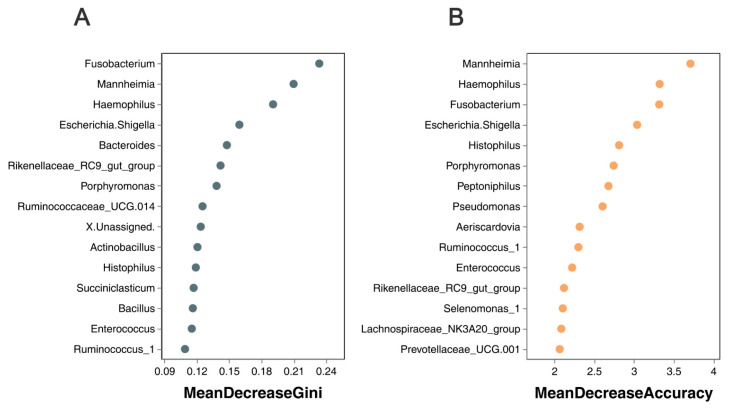
The lung microbiota signature of sheep with varying degrees of pneumonia was determined using Random Forest analysis (95% correct, in genus level). (**A**) The mean decrease Gini. (**B**) The mean decrease accuracy.

## Data Availability

The authors confirm that all data are fully available without restriction. The raw sequencing reads were deposited into the NCBI Sequence Read Archive (SRA) database (Accession Number: PRJNA995047).

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
