# Peer review of "Characterization of Lung Microbiomes in Pneumonic Hu Sheep Using Culture Technique and 16S rRNA Gene Sequencing"

_animals, 2023, doi:10.3390/ani13172763_

Round 1

Reviewer 1 Report

In this study, the Authors combined high-throughput 16SrRNA gene sequencing and bacterial culturing to characterize the viable bacterial community in lung samples from healthy sheep, moderate pneumonia sheep, and severe pneumonia sheep.

The idea and hypothesis are both good, and the topic investigated fits well the overall scope of the journal and the Section.

Moreover, the findings may be useful for veterinary medicine and science.

The Simple Summary section should be reduced.

The Abstract section needs to be revised by adding some numerical results or significance levels.

The Introduction section gives a good overview even if the addition of further recently available literature may add value to the section by adding a couple of addidtional sentences.

Lines 75-81: revise these sentences.

The results have been reported in a clear and simple way.

However, the manuscript needs some revision for English language…some sentences result quite hard to follow...

Further, Figures  should be provided with more high quality.

Check if all the references cited into the text have been reported in the references list.

The Conclusions section should be improved because of in its present form is quite hard to follow.

So, after revision the paper could be accepted for publication.

Moderate editing of English language required

Author Response

Dear reviewer:

  We are very grateful for your comments and thoughtful suggestions. We agree with your suggestions and have modified the terminology throughout the text as appropriate. In addition, we have revised both the Simple Summary section, Abstract section, and Introduction section according to your comments. Detailed revision was shown as follows.

1. The Simple Summary section should be reduced.

Response: We are very grateful for your comments. The reduced Simple Summary section is ‘Pneumonia, which represents a leading cause of mortality among sheep, presents a formidable peril to the sheep industry. In this study, we combine high-throughput 16SrRNA gene sequencing and bacterial culturing to meticulously examine the bacterial community present in lung samples obtained from healthy sheep, those with moderate pneumonia, and those suffering from severe pneumonia. The results demonstrated a significant decrease in pulmonary microbial diversity and alterations in taxonomic composition following the onset of pneumonia. Our results suggest that M. haemolytica and Fusobacterium were the crucial pathogens in the sheep pneumonia. In addition, the occurrence of sepsis in severe pneumonia sheep could contribute to the eventual mortality outcome. Taken together, this study contributes to guiding preventive and therapeutic measures for pneumonia of different severities.’

2. The Abstract section needs to be revised by adding some numerical results or significance levels.

Response: We thank you very much for your comments. We have added the significance levels in the Abstract section. Detailed revision was shown in line 24 to line 30 and marked in red.

3. The Introduction section gives a good overview even if the addition of further recently available literature may add value to the section by adding a couple of additional sentences.

Response: We are very grateful for your comments and suggestions. We added ‘Notably, resistance to tylosin and tetracycline has been observed in a subset of individuals (5)’.

Alvarez J, Calderón Bernal JM, Torre-Fuentes L, Hernández M, Jimenez CEP, Domínguez L, Fernández-Garayzábal JF, Vela AI, Cid D. Antimicrobial Susceptibility and Resistance Mechanisms in Mannheimia haemolytica Isolates from Sheep at Slaughter. Animals. (2023) 13(12), 1991. doi: 10.3390/ani13121991

4. Lines 75-81: revise these sentences. The results have been reported in a clear and simple way. However, the manuscript needs some revision for English language some sentences result quite hard to follow.

Response: We are very grateful for your comments and agreed with your suggestions. The revised sentences are ‘The objective of this study was to characterize the viable bacterial community in lung samples from healthy sheep, moderate pneumonia sheep, and severe pneumonia sheep and to determine the main pathogens. We hope this research will provide insights into prevention and treatment of Hu sheep pneumonia.’

5. Further, Figures should be provided with more high quality.

Response: We thank you very much for your comments for pointing out this omission. The higher quality Figures were listed in the additional files.

6. Check if all the references cited into the text have been reported in the references list. The Conclusions section should be improved because of in its present form is quite hard to follow.

Response: We are very grateful for your comments and agreed with your suggestions. References in this manuscript has been corrected. The rewritten Conclusions section as listed in below.

‘We combine high-throughput 16SrRNA gene sequencing and bacterial culturing to characterize the viable bacterial community in lung samples. The results demonstrated a significant decrease in pulmonary microbial diversity and alterations in taxonomic composition following the onset of pneumonia. M. haemolytica and Fusobacterium were identified as the crucial pathogens in sheep pneumonia. In addition, P. aeruginosa, A. veronii, and P. azotoformans were isolated from blood samples in severe pneumonia sheep (83.3% [5/6]). Thus, the occurrence of sepsis in severe pneumonia sheep could contribute to the eventual mortality outcome. Veterinarians needed to take into account sepsis when develop a treatment plan. Importantly, this study contributes to guiding preventive and therapeutic measures for pneumonia of different severities.’

Reviewer 2 Report

The authors have made a good attempt to explore an interesting area of screening the lung microbiome of healthy and diseased (pneumonia) Sheep using both culture and next generation sequencing based approaches. The results were interesting and so were they discussed fairly well. However there seems to be a lot of typing errors throughout the manuscript, both grammatical and some scientific too. Hence I strongly suggest that the authors go through the entire manuscript thoroughly and rectify for such errors.

There are minor typographic edits to be made, which I suppose would be addressed during the proof reading stage. Some of them have been listed below:

a.       Line 32: “pneumonia. integrating the results of …”; the next sentence is not in sentence case “pneumonia. Integrating the results of…”

b.      Line 15: “e pneumonia. we primarily investigated the” similar to the previously mentioned comment on sentence case

c.       Line 35: change the font style from normal to italics for the word: “Fusobacterium spp”

The authors can check for such minor corrections throughout the text.

Other comments:

1.      Kindly indicate all the organisms in italics font. For e.g., line 42-44: “Currently, Mannheimia haemolytica (M. haemolytica), Pasteurella multocida (P. multocida) are generally recognized as the major 43 causative pathogens of pneumonia in sheep (3, 4).” Indicate the organisms in italics: Mannheimia haemolytica (M. haemolytica), Pasteurella multocida (P. multocida). The names of all organisms should be indicated in italics that is the widely followed scientific writing approach. Kindly follow this strictly throughout the manuscript.

2.      Line 87: “The main clinical presentation is respiratory…” kindly indicate this sentence in past tense: “The main clinical presentation was respiratory…”

3.      Line 95: “… therapy. and the details about…” there’s a typing error here, kindly rectify it

4.      Similarly, on line 105 “Finally, and the whole lungs were”

5.      Line 188-199: Kindly edit this sentence, apart from the fact that its long, there seems to be some words missed as it loses the flow.

6.      Line 213-214: was the word “OUT” typed wrongly? Kindly look into this entire sentence too

7.      Line 225-228: the authors have defined groups H, M and S, if I am correct, this abbreviation was used for the first time here without any prior elaboration. Kindly re-check this. It’s ideal to mention these abbreviations with their respective elaboration in the materials and methods section.

8.      Line 269: the reference indicated was not numbered

9.      Line 327-329: there’s grammatical error in this sentence, punctuation marks were wrongly placed. Kindly rectify this.

10.  Line 414: expand the term ‘ARDS’

There are quite a few typing/grammatical and other language related errors throughout the manuscript. Some of them have been listed in the comments given to the authors. Kindly look into them and revise the manuscript accordingly.

Author Response

Dear reviewer

We are very grateful for your comments and sorry for our carelessness. We agree with your suggestions and modified the terminology throughout the text as appropriate. In addition, we have revised the mistakes according to your comments. The detailed revision is listed as follows.

  1. There are minor typographic edits to be made, which I suppose would be addressed during the proof-reading stage. Some of them have been listed below:

Line 32: “pneumonia. integrating the results of …”; the next sentence is not in sentence case “pneumonia. Integrating the results of…”; Line 15: “e pneumonia. we primarily investigated the” similar to the previously mentioned comment on sentence case; Line 35: change the font style from normal to italics for the word: “Fusobacterium spp”

Response:

We are very grateful for your comments. These errors have been corrected. Minor errors like this are checked throughout the text. Detailed revision is shown in the corrected manuscript marked in red.

  1. Other comments:

(1). Kindly indicate all the organisms in italics font. For e.g., line 42-44: “Currently, Mannheimia haemolytica (M. haemolytica), Pasteurella multocida (P. multocida) are generally recognized as the major 43 causative pathogens of pneumonia in sheep (3, 4).” Indicate the organisms in italics: Mannheimia haemolytica (M. haemolytica), Pasteurella multocida (P. multocida). The names of all organisms should be indicated in italics that is the widely followed scientific writing approach. Kindly follow this strictly throughout the manuscript.

Response:

Thank you for your correction. All the organisms are changed in italic in the corrected manuscript.

(2). Line 87: “The main clinical presentation is respiratory…” kindly indicate this sentence in past tense: “The main clinical presentation was respiratory…”

Response:

We are grateful for your correction. We are sorry for the mistake. Detailed revision is shown in the corrected manuscript marked in red.

(3). Line 95: “… therapy. and the details about…” there’s a typing error here, kindly rectify it.

Response:

We are grateful for your attentive observation. We are sorry for the mistake. Detailed revision is shown in the corrected manuscript marked in red.

(4). Similarly, on line 105 “Finally, and the whole lungs were”

Response:

We are grateful for your attentive observation. We are sorry for the mistake. Detailed revision is “Finally, the separated whole lungs were photographed in a clean background.” shown in the corrected manuscript marked in red.

(5). Line 188-199: Kindly edit this sentence, apart from the fact that its long, there seems to be some words missed as it loses the flow.

Response:

We found that bacteria can be isolated in the lungs from both healthy (50.00% [3/6]) and diseased sheep (100% [12/12]). M. haemolytica is the dominant species isolated from the sheep with moderate pneumonia. However, a greater number of bacterial species and coinfections were found in lungs from severe pneumonia sheep. It is worth noting that five isolates were only recovered in blood from six sheep with severe pneumonia (83.33% [5/6]), including P. aeruginasa (60.00% [3/5]), Aeromonas veronii (A. veronii) (20.00% [1/5]), and Pseudomonas azotoformans (P. azotoformans) (20.00% [1/5]). The sepsis observed in severe pneumonia-afflicted sheep may ultimately contribute to their mortality.

(6). Line 213-214: was the word “OUT” typed wrongly? Kindly look into this entire sentence too

Response:

Thank you for your attentive observation. We are sorry for the typed wrongly mistakes. It is “OTUs”. We have corrected the typed error. Detailed revision is shown in the corrected manuscript marked in red.

(7). Line 225-228: the authors have defined groups H, M and S, if I am correct, this abbreviation was used for the first time here without any prior elaboration. Kindly recheck this. It’s ideal to mention these abbreviations with their respective elaboration in the materials and methods section.

Response:

Thank you for your attentive observation. We fully agree with you and have defined groups H, M and S in the materials. Detailed revision was shown as follows.

The sheep cohort comprised six individuals classified as health (group H), six individuals afflicted with moderate pneumonia but without any therapeutic intervention (group M), and an additional six individuals suffering from severe pneumonia (group S)…

(8). Line 269: the reference indicated was not numbered.

Thank you for your attentive observation. The reference was numbered in 17. Detailed revision is shown in the corrected manuscript, marked in red.

(9). Line 327-329: there’s grammatical error in this sentence, punctuation marks were wrongly placed. Kindly rectify this.

Response:

Thank you for your attentive observation. The rectified sentence was ‘Mannheimia, Haemophilus, Fusobacterium, and Escherichia were identified as the most discriminating predictors based on the mean decrease in accuracy (Figure 8B)’. Detailed revision is shown in the corrected manuscript marked in red.

(10). Line 414: expand the term ‘ARDS’

Response:

We are very grateful for your comments and suggestions. We expanded the term ‘ARDS’ as acute respiratory distress syndrome. Detailed revision is shown in the corrected manuscript marked in red.

Round 2

Reviewer 1 Report

The revised paper merits the final acceptance.